# One Training Fits All: Addressing Model-Heterogeneity Federated Learning via Architecture Probing

## Abstract

Model-heterogeneity federated learning (FL) is a flexible setting where a client trains a model subject to its local computation capacity. Towards the scenario, partial averaging extracts the clients' models from a global model so that the aggregation of each model parameter is identified. While existing models can only generate submodels with predefined settings established during training, our approach utilizes a trainable probabilistic masking strategy named FedMAP, enabling the dynamic creation of customized model sizes aligned with the client's budget. In detail, the clients find the best model architectures based on their local datasets and computation resources, and the FL server merges these local optimal architectures into a probabilistic mask. In the end, we attain a stable probabilistic mask, with which we can generate arbitrary models for evaluation or update the counterpart of the model parameters while training with the clients' data. Our experiments validate the effectiveness of the proposed FedMAP from two aspects: (i) It can improve the state-of-the-art approaches to heterogeneous model updates, especially for those small-size models; and (ii) We can extract the submodels whose sizes never appear in training with exceptional performance.

## 1 Introduction

Federated learning (FL) (Konečnỳ et al., 2016; McMahan et al., 2017) is conceptualized to facilitate inter-client collaboration where a number of edge devices (e.g., smartphones) jointly train a global model under the orchestration of a central server. Thanks to its distributed nature that enhances both data and computation utilization, FL has been widely deployed as an efficient solution in various commercial products (He et al., 2020; Beutel et al., 2020; Lai et al., 2022; Xie et al., 2023). However, when scaled up to a vast number of client devices, FL is confronted with the challenge of computation heterogeneity, where the computation capacities of the clients are considerably different from each other.

To address this challenge, a common approach is to enable model heterogeneity, ensuring that the model deployed on each individual client align with its local computation capacity. This can be done by extracting a submodel for each client from a global model, which encompasses a subset of the parameters of the global model. Existing works to extract heterogeneous submodels can be categorized in two different groups, namely, static submodel extraction (e.g., HeteroFL (Diao et al., 2020) and DepthFL (Kim et al., 2022)) and rolling-based submodel extraction (e.g., FedRolex (Alam et al., 2022)). As illustrated in Figure 1, both submodel extraction methods construct the submodels by selecting some neurons at each neural network layer and connecting the neurons between two consecutive layers at the beginning of the model training. During the model training, the static submodel extraction keeps the initial submodel architectures unchanged. In contrast, the rolling-based submodel extraction employs a sliding window to select the neurons and rolls the sliding window during the training process, changing the selected neurons and rebuilding the submodels at every communication round.

While the above existing works are capable of extracting submodels with appropriate sizes, the performance of extracted submodels depend highly on their architectures, and improper choices of neurons often make them suffer from accuracy degradation (Qu et al., 2022; Isik et al., 2022; Alam

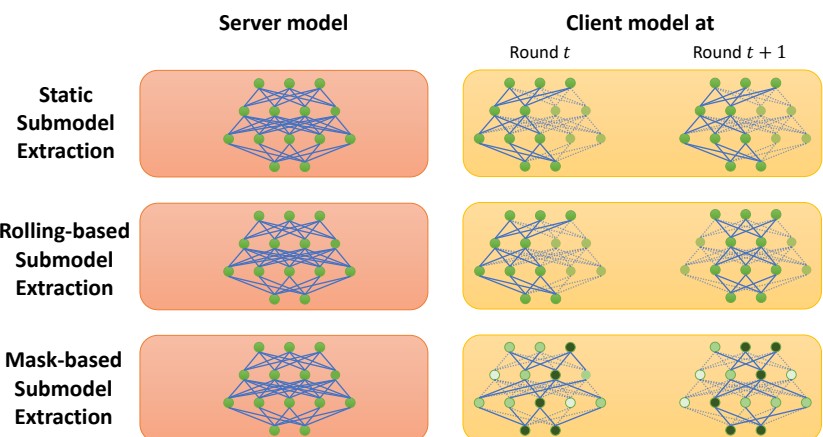

Figure 1: Three types of heterogeneous models' extraction. The first two rows come from the existing works, i.e., static and rolling-based submodel extractions. As for mask-based submodel extraction (our proposed method), there are three different levels of nodes. Neurons with a light green color do not have any connections with others; neurons with a dark color have connections with all neighboring neurons except the light one; otherwise, the connection depends on the computation capacity.

et al., 2022; Liao et al., 2023). Unfortunately, there is no formal analysis on the influence of submodel architectures in the literature, impeding one from defining and studying its optimality. To fill the gap, this study builds a more rigorous understanding of the influence of submodel architectures. Our theoretical insights reveal that existing submodel extraction strategies are unflexible and unlikely to reach the global optimal solution. Moreover, their inflexibility becomes apparent during the testing phase because they are constrained by a predetermined set of submodels established during training. This lack of adaptability becomes especially problematic when new devices come with computational capabilities that are not considered during the initial training. More specifically, the limited number of predetermined submodels can result in impossibility of accommodating devices with lower capacity than any of the clients involved in the training phase and can also lead to inefficient utilization of new devices' computation resources if they are more computationally capable than existing clients.

With the goal of overcoming these limitations, we introduce a novel model heterogeneity aware federated learning framework, named **Fed**erated learning with **M**asked **A**rchitecture **P**robing (FedMAP), which is able to probe the optimal submodel architectures and extract submodels of any size. As illustrated in Fig. 1, the proposed FedMAP can identify the importance levels of neurons by means of a probabilistic mask, and filter out unnecessary neurons while preserving the most important part of the submodel. To further improve the performance of model-heterogeneity FL, we develop a coordinate descent algorithm, which alternatively optimizes the probabilistic mask and the model parameters using FedMAP and the existing partial averaging algorithms (Diao et al., 2020; Horvath et al., 2021), respectively.

Compared to previous works, there are two key advantages of the proposed FedMAP. First, it explicitly optimizes submodel architectures, thereby mitigating the performance degradation due to the use of sub-optimal architecture. Second, different from existing submodel extraction works, our method possesses the capability to generate a new submodel in accordance with a newly arrived client whose resource budget differs from existing clients, thereby unlocking new possibilities and accommodating various computational capacities. This flexibility enables the adaptation to environments with varied resource requirements, ensuring optimal utilization and avoiding resource waste, thereby addressing the limitations of predefined submodel extraction strategies.

**Contribution.** Throughout the paper, our contributions are highlighted as follows:

- To the best of our knowledge, we provide the first theoretical analysis on the convergence of federated learning algorithms under model heterogeneity. Our analysis reveals limitations of prior

methods in identifying optimal architectures in this setting. To tackle this limitation, we jointly optimize architecture and parameters to unlock performance gains.

- Built upon our theoretical analysis, we derive a probabilistic-based solution that learns a mask enabling customized submodel generation according to clients' resource budgets. This allows training the framework once, then efficiently deploying optimized submodels for heterogeneous edge devices by sampling important parameters based on their capacities. Our mask learning approach thus facilitates a flexible one-training-fits-all solution while accounting for device variability.

- Extensive experiments show FedMAP significantly outperforms existing approaches, particularly for low-resource clients. Pronounced gains on resource-limited devices demonstrate FedMAP's strength in efficiently tailoring submodels to diverse capabilities.

## 2 RELATED WORKS

**Heterogeneous Model Aggregation.** Training with the heterogeneous clients' models provides flexibility in handling computation heterogeneity. There are two different ways to achieve aggregation among heterogeneous models: knowledge distillation (Lin et al., 2020; Zhang et al., 2021; Itahara et al., 2021; Cho et al., 2022) and partial averaging (Diao et al., 2020; Horvath et al., 2021; Alam et al., 2022; Kim et al., 2022; Zhu et al., 2022; Ilhan et al., 2023; Liao et al., 2023). Knowledge distillation achieves heterogeneous aggregation via logits alignment, while partial averaging maps the parameters to the global model and updates the counterpart. Although knowledge distillation effectively transfers knowledge across all clients, it is infeasible due to the necessity of a public dataset. Although our proposed method uses the partial averaging as well, we explore the optimal submodel architectures when compared with the existing works, as discussed in Section 1.

**Other solutions to computation heterogeneity in FL.** In contrast to the above method, there is another type of work for computation heterogeneity with the goal of achieving synchronous global aggregation, making no extra waiting for the faster devices. In specific, the algorithms allow the clients to perform multiple local updates before the model aggregation on the server, where the numbers of local updates vary among the clients (Wang et al., 2020; Li et al., 2020b; Mitra et al., 2021; Luo et al., 2021; Shin et al., 2022; Wu et al., 2023). These works require the clients to fully train the model. When a global model is very large, these works are no longer feasible because the clients can hardly load the model.

**Model Pruning in Federated Learning.** Meanwhile, we review some computation-efficient approaches in FL, such as model pruning, which are related to our algorithm design. An emerging interest has arisen in the pursuit of discovering sparse networks through pruning, with the potential to achieve remarkably high computation efficiency while maintaining performance in terms of accuracy. Though pruning is widely used (Zhou et al., 2019; Li et al., 2020a; 2021; Isik et al., 2022), these works come from the perspective of saving the computation cost but neglect the various constraints on the computation resources of distinct devices.

## 3 PRELIMINARY AND THEORETICAL ANALYSIS ON MODEL ARCHITECTURE

Given an FL system consists of $N$ clients, each has computation capacity $\gamma_i \in [0,1], \forall\, i \in [N]$. Model-heterogeneity FL (Diao et al., 2020; Horvath et al., 2021; Alam et al., 2022) explicitly associates client $i$ with a binary-valued mask $\mathcal{M}_i$ on the model parameter $\tilde{\boldsymbol{x}} \in \mathbb{R}^d$ such that at most $\gamma_i d$ parameters are learnt on it and solves the following objective

$$\min_{\tilde{\boldsymbol{x}} \in \mathbb{R}^d} F(\tilde{\boldsymbol{x}}, \mathcal{M}) = \frac{1}{N} \sum_{i \in [N]} F_i(\tilde{\boldsymbol{x}} \odot \mathcal{M}_i), \tag{1}$$

where $\mathcal{M} \in \{0,1\}^{N \times d}$ denotes the collection of all masks and the capacity constraint requires that $\|\mathcal{M}_i\|_1 \leq \gamma_i d$. We assume that $\mathcal{M}$ is generated from some distribution $P$, which encompass deterministic masking strategies by treating any of them as sampling a mask with probability 1. In addition, $F_i(\cdot)$ is the loss of client $i$ with respect to its given parameters $\tilde{\boldsymbol{x}} \odot \mathcal{M}_i$. To be more specific, $F_i(\tilde{\boldsymbol{x}}) = \frac{1}{|\mathcal{D}_i|} \sum_{(d_{in}, d_{out}) \in \mathcal{D}_i} \mathcal{L}(\tilde{\boldsymbol{x}}; d_{in}, d_{out})$, where $\mathcal{D}_i$ is the dataset of client $i$, and $\mathcal{L}(\cdot; \cdot, \cdot)$ is the loss function with respect to the given parameters and the training data.

With the above problem formulation, we use the following case study to understand how to determine if the model converges in a non-convex case for a given mask.

**Case study: Quadratic functions.** Assume $F_i(\tilde{\boldsymbol{x}}) = \frac{1}{2} \left\| \left(\tilde{\boldsymbol{x}} - \boldsymbol{x}_*^{(i)}\right) \odot \mathcal{M}_i \right\|_2^2$ and $\omega_i = 1/N$ for all $i \in [N]$. Therefore, the optimal solution to $F(\tilde{\boldsymbol{x}})$ is

$$\boldsymbol{x}_* = \cup_j (\boldsymbol{x}_*)_j, \quad (\boldsymbol{x}_*)_j = \frac{1}{\left(\sum_{i \in [N]} \mathcal{M}_{ij}\right)} \sum_{i \in [N]} \left(\boldsymbol{x}_*^{(i)}\right)_j \cdot \mathcal{M}_{ij}.$$

For simplicity, we denote the result of $\boldsymbol{x}_*$ by $Agg_{i \in [N]}(\boldsymbol{x}_*^{(i)} \odot \mathcal{M}_i)$. Therefore, in the non-convex objectives, we aim to show that $Agg_{i \in [N]}(\nabla F_i(\tilde{\boldsymbol{x}}_t \odot \mathcal{M}_i) \odot \mathcal{M}_i)$ can converge to 0 for $t \in \{1, 2, \dots\}$. In the following analysis, we denote $Agg_{i \in [N]}(\nabla F_i(\tilde{\boldsymbol{x}}_t \odot \mathcal{M}_i) \odot \mathcal{M}_i)$ by $\nabla F(\tilde{\boldsymbol{x}}_t)$.

**Partial Averaging.** With reference to Figure 1, static submodel extraction optimizes $\tilde{\boldsymbol{x}}$ in accordance with the constant masks $\mathcal{M}_i$ for all clients $i \in [N]$, and rolling-based submodel extraction has limited options for $\mathcal{M}_i$ and changes over the communication rounds. The steps of partial averaging for a general case are described below: At Round $t \in \{0, 1, \dots\}$,

- **Sampling:** The server randomly samples a subset of clients $\mathcal{A} \subset [N]$ and distributes the global model parameters $\tilde{\boldsymbol{x}}_t$ to the selected clients

- **Local Model Training:** The clients $i \in \mathcal{A}$ performs $K$-times local updates via $\boldsymbol{x}_{t,k}^{(i)} = \boldsymbol{x}_{t,k-1}^{(i)} - \eta \nabla F_i \left(\boldsymbol{x}_{t,k-1}^{(i)} \odot \mathcal{M}_i^t\right) \odot \mathcal{M}_i^t$, where $k \in \{1, \dots, K\}$ and $\boldsymbol{x}_{t,0}^{(i)} = \tilde{\boldsymbol{x}}_t$.

- **Global Model Aggregation:** The clients collect the model updates from the participants $\mathcal{A}$ and perform the global model aggregation via $\tilde{\boldsymbol{x}}_{t+1} = \tilde{\boldsymbol{x}}_t - \eta_s Agg_{i \in \mathcal{A}} \left(\boldsymbol{x}_{t,K}^{(i)} - \tilde{\boldsymbol{x}}_t\right)$.

To analyze how the mask affects the model training, we work with the following assumptions:

**Assumption 3.1** (Masked-L-smoothness). Given a fixed binary mask $\mathcal{M} \in \{0,1\}^d$, the gradient of a function $F_i$ is $L$-Lipschitz continuous, i.e., $\|\nabla F_i(w \odot \mathcal{M}) - \nabla F_i(v \odot \mathcal{M})\| \leq L\|(w-v) \odot \mathcal{M}\|$ for any $w, v \in \mathbb{R}^d$.

**Assumption 3.2** (Masked Bounded Gradient). Given the arbitrary mask $\mathcal{M}_i \in \{0,1\}^d$ and the arbitrary vector $w \in \mathbb{R}^d$, the gradient of a function $F_i$ is bounded by a scalar $G \geq 0$ for all $i \in [M]$, i.e., $\|\nabla F_i(w \odot \mathcal{M}_i)\|_2^2 \leq G^2$

As for the mask is set to be $\{1\}^d$ in the above assumptions, they are reduced to the assumptions that are widely used in federated learning research (Yang et al., 2020; Karimireddy et al., 2020). Based on the assumptions above, we can obtain the following theorem:

**Theorem 3.3.** *Suppose the global optimal result for Eq. (1) is $F_* := F(\boldsymbol{x}_*, \mathcal{M}_*)$, but both values (i.e., $\boldsymbol{x}_*$ and $\mathcal{M}_*$) are unattainable by the clients. Suppose that Assumption 3.1 and 3.2 hold. At $t$-th round, the mask is $\mathcal{M}^t \in \{0,1\}^{N \times d}$. Let $N_j = \sum_{i \in [N]} (\mathcal{M}_i^t)_j$. We set $\eta_s$ as a vector, which can change over the round, where at the $j$-th element of $\eta_s$ should be*

$$(\eta_s)_j = \frac{\binom{N}{A}}{\binom{N}{A} - \binom{N-N_j}{A}}. \tag{2}$$

*We assume the local learning rate $\eta$ is small enough. Therefore, when the model is updated with partial averaging, the loss between two consecutive communication rounds follows an update that*

$$\mathbb{E}F(\tilde{\boldsymbol{x}}_{t+1}, \mathcal{M}_*) \leq F(\tilde{\boldsymbol{x}}_t, \mathcal{M}_*) - \frac{\eta K}{2} \|\nabla F(\tilde{\boldsymbol{x}}_t, \mathcal{M}_*)\|_2^2 + O\left(\eta^2 K^2 G^2\right)$$
$$+ \eta K \mathbb{E}_{\mathcal{M}_t \sim \mathcal{P}} \left\| \left(\nabla F(\tilde{\boldsymbol{x}}_t, \mathcal{M}^t) - \nabla F(\tilde{\boldsymbol{x}}_t, \mathcal{M}_*)\right) \cdot \mathbf{1}_{\{\nabla F(\tilde{\boldsymbol{x}}_t, \mathcal{M}_*) \neq 0\}} \right\|_2^2 \tag{3}$$

The theorem gives an intuition that whether the loss gets decreased depends on the differences between the mask and the optimal mask. When the fixed mask is consistent with the optimal one, we have the following decent corollary for the convergence guarantee:

**Corollary 3.4.** *Suppose the optimal mask keeps consistent with the mask that is used for the model updates. Under the setting of Theorem 3.4, partial averaging will achieve a convergence rate of*

$$\min_{t \in [T]} \left\| \nabla F(\tilde{\boldsymbol{x}}_t, \mathcal{M}^t) \right\|_2^2 \leq O\left( \frac{F(\tilde{\boldsymbol{x}}_0, \mathcal{M}^0) - F_*}{\sqrt{T}} \right) + O\left( \frac{G^2}{\sqrt{T}} \right). \tag{4}$$

*Remark* 3.5. According to the above corollary, the convergence rate for partial averaging is $O\left(1/\sqrt{T}\right)$, which is consistent with the state-of-the-art algorithms. This corollary indicates that partial averaging can definitely converge to surrogate results when the mask is fixed. However, another insight from the theorem is that partial averaging is unlikely to achieve the global optimal results because its gradient on the optimal mask is still greater than 0 when the fixed and the optimal masks mismatch. Motivated by the theoretical findings, it is not trivial to find the optimal mask, which will be proposed in the next section. With the best mask we can find, we update the model parameters via the existing partial averaging algorithms. In case the new model is yet to achieve the best performance, we can follow the idea of coordinate descent algorithm (Wright, 2015) to realize an algorithm that can alternatively update the probabilistic mask and the model parameters. Further details will be discussed in the next section.

## 4    FEDMAP: AN APPROACH FOR MASKED-BASED SUBMODELS EXTRACTION

Section 3 discusses the importance of finding an initial binary mask, which relies on a predefined distribution. Therefore, it is important to discuss how to find the distribution such that a proper binary distribution mask can be drawn. In this section, we introduce a trainable probabilistic masking strategy named FedMAP with a focus on how the clients train the mask and how the server aggregates the mask. A detailed algorithm pseudocode is described in Algorithm 1.

### 4.1    PROBLEM FORMULATION AND MASK GENERATION RULES

Binary masks $\{\mathcal{M}_i\}_{i \in [N]}$ involve discrete variables which make the global optimum NP-hard to achieve (Pensia et al., 2020). Inspired by (Rolfe, 2016), we instead consider that each $\mathcal{M}_i$ follows distribution $f(\mathcal{M}_i \mid p; \gamma_i)$ parameterized by a client-independent trainable $p$ and client-dependent capacity $\gamma_i$. We further assume that $f(p)$, the prior of $p$, is a factorized Beta distribution, i.e., $p \in [0, 1]^d$ has $d$ independent components, each following a univariate Beta distribution. For simplicity, we denote $f(p) = \text{Beta}(\alpha, \beta)$ where $\alpha, \beta \in \mathbb{R}^d$. Put together, we solve the following optimization in lieu of Eq. (1)

$$\min_{\tilde{\boldsymbol{x}}, \alpha, \beta} \mathbb{E}_{\mathcal{M}, p}[F(\tilde{\boldsymbol{x}}, \mathcal{M})], \quad \text{s.t. } \mathcal{M} = \{\mathcal{M}_i\}_{i \in [N]}, \ \mathcal{M}_i \sim f(\mathcal{M}_i \mid p; \gamma_i), \ p \sim \text{Beta}(\alpha, \beta). \tag{5}$$

The merit of Eq. (5) lies in twofold: First, it establishes a quasi-Bayesian framework where likelihood $f(\mathcal{M}_i \mid p; \gamma_i)$ behaves more alike a factorized Bernoulli distribution $\text{Bern}(p)$ (defined similarly to the factorized Beta distribution) which conjugate with $f(p)$ as $\gamma \to 1$. This allows us to take insights from previous works on heterogeneous mask aggregation (Zhou et al., 2019; Isik et al., 2022) and derive an efficient aggregation algorithm as detailed shortly. Second, this formulation incorporates randomness in mask learning and helps escape from poor local optimums, often resulting in further empirical improvement. In the remaining part of this section, we present an effective way to solve Eq. (5) approximately.

**Solution Overview.**    To solve non-convex Eq. (5), We opt for coordinate descent by learning $\tilde{\boldsymbol{x}}$ and $\alpha, \beta$ alternatively. The update of $\tilde{\boldsymbol{x}}$ given $\alpha, \beta$ follows the conventional partial averaging approaches (e.g., HeteroFL (Diao et al., 2020), FjORD (Horvath et al., 2021)), and therefore, we omit the detailed steps in the main context, but we provide a pseudocode in Algo. 3. We apply Algo. 3 in our experiments. In terms of optimizing $\alpha, \beta$, we resort to straight-through estimator (Bengio et al., 2013) to optimize $F_i$ with respect to $p$ on client $i$ separately. Notably, each $p$ learned on client $i$ is used as a proxy of the posterior of $\mathcal{M}_i$ and we aggregate all of them to update the posterior of $p$ in the server.

## 4.2 Learning

**Sampling Distribution of Masks.** Mask $\mathcal{M}_i \sim f(\mathcal{M}_i \mid p, \gamma_i)$ for client $i \in [N]$ is generated and subjected to constraints $\gamma_i$. To this end, one way is to employ rejection sampling, i.e., keep drawing $\mathcal{M}_i \sim \text{Bern}(p)$ until $\|\mathcal{M}_i\|_1 \leq \gamma_i d$. However, this may encounter long wait time for small $\gamma_i$. For speedup, we instead set the candidate set for unmasked positions in each round to the outcome of the the previous round. In particular, we initialize the candidate mask as $\bar{\mathcal{M}}_i^0 = \mathbf{1}^d$, and at $\kappa$-th round, a new candidate is drawn $\bar{\mathcal{M}}_i^\kappa \sim \text{Bern}(p \odot \bar{\mathcal{M}}_i^{\kappa-1})$ until $\|\bar{\mathcal{M}}_i^\kappa\|_1 \leq \gamma_i d$.

**Local Updates of Masks.** Suppose client $i \in [N]$ holds a model of $\tilde{x}$ and the probabilistic mask $p$. While optimizing $p$, the client should guarantee the training cost within the computation constraint $\gamma_i$. Now that we find a submodel that satisfies the computation constraints, we discuss how the value of $p$ updates throughout the local training process. There are two key challenges when optimizing the value of $p$ with the traditional SGD approach. Firstly, The values of $p$ are in the domain of $[0, 1]$, which are hard to control during the local training. Secondly, the proposed mask generation strategy follows a discrete probability distribution, meaning that it is not differentiable over the value of $p$, so we cannot directly use SGD to update $p$. To alleviate these two challenges, we project the domain of $p$ to a real-number vector $\mathbb{R}^d$ and adopt the straight-through estimator (Bengio et al., 2013) to optimize $p$. The following proposition gives a detailed description of how $p$ updates, and the justification is given in Appendix B.

**Proposition 4.1.** *For probabilistic mask $p \in [0, 1]^d$, we transform it to an unconstrained vector $s \in \mathbb{R}^d$ by applying the inverse of the sigmoid function $\sigma(\cdot)$ elementwisely, i.e., $s = \sigma^{-1}(p)$. Client $i \in [N]$ has expected loss $F_i$ and the maximum computation capacity $\gamma_i$, for which a mask is sampled $\mathcal{M}_i \sim f(\mathcal{M}_i|p, \gamma_i)$. We define vector $m \in \mathbb{Z}_+^d$ upon the candidate mask sequences $\{\bar{\mathcal{M}}_i^\kappa\}_{\kappa=1}^\infty$ by*

$$m = \sum_{\kappa=1}^{\min\{\kappa:\|\bar{\mathcal{M}}_i^\kappa\|_1 \leq \gamma_i d\}} \bar{\mathcal{M}}_i^\kappa. \tag{6}$$

*In words, $m$ counts for each position how many times 1 is sampled until the valid mask $\mathcal{M}_i$ is generated. Suppose the client holds the global model parameters $\tilde{x}$, we update the probabilistic mask $p$ through the following steps:*

- ***Shrink the mask** $\mathcal{M}_i$: The above mask guarantees the forward propagation within the computation constraint $\gamma_i$, but the backward propagation is likely to exceed the constraint. Therefore, we shrink the mask smaller to ensure the backward propagation also within the computation constraint. Therefore, we take one more Bernoulli step and update the mask $\mathcal{M}_i$ and $m$ via: (i) $\mathcal{M}_i \sim Beta(p_i \odot \mathcal{M}_i)$, and (ii) $m \leftarrow m + \mathcal{M}_i$.*

- ***Update transformed** $s$: We follow an SGD step via $s \leftarrow s - \eta_m \nabla_s F_i(\tilde{x} \odot \mathcal{M}_i)$, where $\eta_m$ is the learning rate, and*

$$\nabla_s F_i(\tilde{x} \odot \mathcal{M}_i) = \nabla F_i(\tilde{x} \odot \mathcal{M}_i) \cdot \tilde{x} \cdot \mathbf{1}_{\{m \geq \|m\|_\infty - 1\}} \left( \frac{p^2 - p^{2m+2}}{1+p} \right) \tag{7}$$

*where $\|m\|_\infty$ returns the maximal entry of $m$.*

- ***Update the probabilistic mask** $p$: We recover updated $p \leftarrow \sigma(s)$.*

**Case study on Quasi-Bayesian Updates on equal $\gamma$.** Let us consider a special case where all clients have the equivalent computation capacity, i.e., $\gamma_i = \gamma$ for all $i \in [N]$. Then, the probabilistic mask is thereby generated from $p \sim \text{Beta}(\alpha, \beta|\gamma)$. Client $i$ can get the value of $m$ from Eq. (6) while generating the mask $\mathcal{M}_i \sim f(\mathcal{M}_i|p_i, \gamma)$, and therefore, $m$ can be regarded as drawing from a distribution $g(m|p_i, \gamma)$, where, at $j$-th element ($j \in [d]$),

$$g(m_j|p_{ij}) = \begin{cases} p_{ij}^{m_j}(1-p_{ij}), & m_j < \|m\|_\infty \\ p_{ij}^{m_j}, & m_j = \|m\|_\infty \end{cases} \tag{8}$$

We denote the above $m$ as $m_i$ for client $i$. Apparently, Bata distribution is a conjugate prior in this case. Therefore, as long as the clients $i \in \mathcal{A}$ transmit $m_i$ to the server, where $\mathcal{A}$ is a set of the

---

**Algorithm 1** FedMAP

---

**Input:** masking learning rate $\eta_m$, local updates $K$, Reset Prior $\tau$, Bata distribution parameters $(\alpha_0, \beta_0)$, initial model $\tilde{x}_0$, total communication rounds $T$.

1: Communicate the initial model $\tilde{x}_0$ to all clients $i \in [N]$
2: **for** $t = 0, 1, 2, \ldots, T-1$ **do**
3:      Sample clients $\mathcal{A} \subseteq [N]$
4:      Communicate the probabilistic mask $p_t = \frac{\alpha_t - 1}{\alpha_t + \beta_t - 2}$ with clients $i \in \mathcal{A}$
5:      **for** $i \in \mathcal{A}$ **in parallel do**
6:          Compute $s_{t,0}^{(i)} = \sigma^{-1}(p_t)$
7:          **for** $k = 0, \ldots, K-1$ **do**
8:              $\mathcal{M}_{i,\_} = \mathsf{MaskGeneration}\left(\sigma\left(s_{t,k}^{(i)}\right), \gamma_i\right)$      $\triangleright$ Algo. 2 defines $\mathsf{MaskGeneration}$
9:              $s_{t,k+1}^{(i)} = s_{t,k}^{(i)} - \eta_m \nabla_{s_{t,k}^{(i)}} F_i(\tilde{x}_0 \odot \mathcal{M}_i)$
10:          **end for**
11:          $\_, m_i = \mathsf{MaskGeneration}\left(\sigma\left(s_{t,K}^{(i)}\right), \gamma_i\right)$
12:          Communicate $m_i$ with the server
13:      **end for**
14:      **if** $t \% \tau == 0$ **then**
15:          $\alpha_t = \alpha_0, \quad \beta_t = \beta_0$
16:      **end if**
17:      $\alpha_{t+1} = \alpha_t + \sum_{i \in \mathcal{A}} m_i, \quad \beta_{t+1} = \beta_t + |\mathcal{A}| \cdot \mathbf{1} - \sum_{i \in \mathcal{A}} \mathbf{1}_{\{m_i = \|m_i\|_\infty\}}$
18: **end for**
19: **return** $(\alpha_T - 1)/(\alpha_T + \beta_T - 2)$

---

selected clients, the server can aggregate and update the prior knowledge via

$$\alpha = \alpha + \sum_{i \in \mathcal{A}} m_i, \qquad \beta = \beta + |\mathcal{A}| \cdot \mathbf{1} - \sum_{i \in \mathcal{A}} \mathbf{1}_{\{m_i = \|m_i\|_\infty\}} \tag{9}$$

### 4.3 Algorithm Design

**Initialization and server setup (Line 3 – 4).** We initialize hyper-parameters $\alpha_0, \beta_0 \in \mathbb{R}^d$ in Beta distribution for $\alpha_0 = \beta_0 = \lambda \cdot \mathbf{1}$ ($\lambda > 1$) to learn as follows. As suggested by (Ferreira et al., 2021; Isik et al., 2022), in each training round, the probabilistic mask $p$ should be set as the maximum likelihood estimators, i.e., $p = \frac{\alpha - 1}{\alpha + \beta - 2}$. It is universally acknowledged that the changes in model parameters likely lead to different optimal masks. Therefore, we fix the model parameters $\tilde{x}$ during the update of the probabilistic mask $p$, with which we can generate a set of optimal masks $\mathcal{M}$ for all clients. In other words, when the server initiates a new round for probabilistic mask optimization, it only transmits the probabilistic mask $p$ to the selected clients if they have held $\tilde{x}$.

**Local updates on clients (Line 5 – 13).** Suppose the clients hold the global model $\tilde{x}$. After receiving the probabilistic mask $p$ from the server, the clients should start training the probabilistic mask locally. We let $p_t \in [0,1]^d$ be the initial probabilistic mask received from the server at round $t \in \{0, 1, ...\}$. We repeat the steps in Proposition 4.1 for $K$ times and obtain the updated probabilistic mask $\tilde{p}_t^i$ on client $i$. As the probabilistic mask generated by the server is based on Beta distribution, the aggregation should be built upon its parameters $(\alpha, \beta)$. In this case, the client should transmit how to update $\alpha'$ and $\beta'$ to the server, which depends on $\mathcal{M}_i \sim f(\mathcal{M}_i | p_i, \gamma_i)$, and $\alpha' = m$ as defined in Eq. (6).

**Server aggregation (Line 14 – 17).** After receiving the masks from the selected clients, i.e., $\{m_i\}_{i \in \mathcal{A}}$, we take Eq. (9) to update $(\alpha, \beta)$. However, these two parameters keep increasing during the training, which probably harms the performance. To add some perturbation and give higher confidence to update the probabilistic mask value, we reset these two parameters from time to time. We have analyzed the way under a special case where all clients have the same computation capacity, which inspires us to work with it under heterogeneous settings. The proposed way can still apply to the case where the clients have heterogeneous computation capacity. The values on each weight

| Datasets | Submodel Extraction | HeteroFL | | | | FjORD | | | |
|---|---|---|---|---|---|---|---|---|---|
| | | 1.0 | 1./4 | 1./16 | 1./64 | 1.0 | 1./4 | 1./16 | 1./64 |
| CINIC-10 | Static | 34.22 | 32.49 | 31.39 | 29.18 | 35.02 | 38.82 | 39.42 | 34.14 |
| | FedRolex | 39.21 | 36.49 | 31.48 | 28.12 | NA | NA | NA | NA |
| | FedMAP | **39.95** | **39.46** | **37.33** | **33.32** | **40.26** | **42.22** | **40.51** | **35.1**2 |
| CIFAR-100 | Static | 24.36 | 22.63 | 20.28 | 17.30 | 31.85 | 33.01 | 32.72 | 32.40 |
| | FedRolex | **30.17** | 28.18 | 21.58 | 11.71 | NA | NA | NA | NA |
| | FedMAP | 27.73 | **30.95** | **27.92** | **22.89** | **33.20** | **34.89** | **35.75** | **35.44** |

Table 1: Comparison among baselines in terms of the final accuracy (%) after 1000 rounds in CINIC-10 and CIFAR-100. **Bold**: The best result in each column for each dataset.

| Datasets | Submodel Extraction | HeteroFL | | | | FjORD | | | |
|---|---|---|---|---|---|---|---|---|---|
| | | 1.0 | 0.64 | 0.36 | 0.16 | 1.0 | 0.64 | 0.36 | 0.16 |
| Shakespeare | Static | 43.57 | 44.63 | 36.89 | 36.53 | 42.65 | 42.69 | 42.52 | 42.37 |
| | FedRolex | 45.83 | 45.01 | 43.56 | 37.7 | NA | NA | NA | NA |
| | FedMAP | **46.61** | **46.41** | **45.64** | **43.68** | **48.63** | **48.55** | **47.03** | **44.73** |

Table 2: Comparison among baselines in terms of the final accuracy (%) after 1000 rounds in Shakespeare. **Bold**: The best result in each column for each dataset.

reflect the importance of each parameter. In this means, we are likely to eventually achieve $\mathcal{M}_0 \subset \mathcal{M}_1 \subset \cdots \subset \mathcal{M}_{N-1}$ as $\gamma_0 \leq \gamma_1 \leq \cdots \leq \gamma_N$. When we try to extract a submodel somewhat in the middle of two consecutive training model sizes, it can perform better than the smaller size.

## 5 EXPERIMENTS

**Datasets and Models.** We evaluate the proposed method with two computer vision (CV) datasets and one natural language processing (NLP) dataset, namely, CINIC-10 (Darlow et al., 2018) and CIFAR-100 (Krizhevsky et al., 2009) for image classification, and Shakespeare (McMahan et al., 2017) for next character prediction. For the first two datasets, we train a ResNet-18 (He et al., 2016) and replace its batch normalization (BN) layers with static BN ones (Diao et al., 2020). For Shakespeare, we train a 2-layer LSTM (Reddi et al., 2020).

**Data Heterogeneity.** We follow (Caldas et al., 2018; He et al., 2020) and partition Shakespeare such that it preserves the non-i.i.d. features with a total of 715 clients, where each client holds inconsistent numbers of training samples. For CIFAR-100 and CINIC-10, we follow (Hsu et al., 2019; Jhunjhunwala et al., 2022) and partition the datasets into 100 and 200 clients, respectively, based on a Dirichlet distribution setting $\alpha = 0.3$.

**Baselines.** In this section, we consider two partial averaging approaches, namely, HeteroFL (Diao et al., 2020) and FjORD (Horvath et al., 2021), and two submodel extractions, i.e., static submodel extraction (Diao et al., 2020) and rolling-based submodel extraction (or FedRolex (Alam et al., 2022)). As mentioned by Alam et al. (2022), FedRolex is not compatible with FjORD.

**System Heterogeneity and Implementation.** For CINIC-10 and CIFAR-100, we consider four different types of clients hold $\{1, 1/4, 1/16, 1/64\}$ of the full model's parameters. As for the Shakespeare dataset, we also consider four types of clients but with different model settings, i.e., $\{1.0, 0.64, 0.36, 0.16\}$. To avoid the randomness of the results, we averaged the results from three different random seeds. In the experiments, we report the results of all the baselines based on the best hyperparameter settings.

### 5.1 TEST ACCURACY OVER DIFFERENT TASKS

In this section, we focus on the empirical results of three tasks in Table 1 and 2. These two tables show the test accuracy against different model sizes for different partial averaging approaches. Thus,

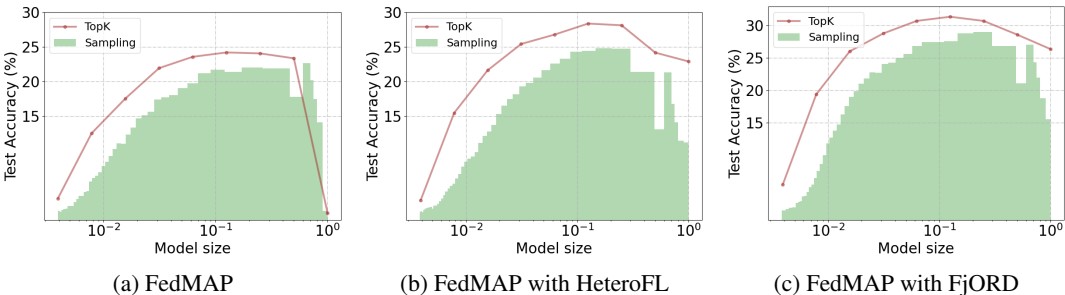

|                          |                                  |                               |
| :----------------------: | :------------------------------: | :---------------------------: |
|       (a) FedMAP         |     (b) FedMAP with HeteroFL     |     (c) FedMAP with FjORD      |

Figure 2: Comparison of different submodel extraction strategies based on the proposed FedMAP using test accuracy against the model sizes after 1000 communication rounds when we train ResNet-18 using CIFAR-100.

we discuss our proposed submodel extraction (FedMAP) among different tasks and different partial averaging approaches, followed by a comparison with the existing submodel extraction approaches.

**The influence of the model sizes on FedMAP.** Interestingly, the best performance of our proposed submodel extraction is likely not at the full model size in the CV datasets. As presented in Table 1, the models with $1./4$ and $1./16$ of the parameters are likely to have a better performance than the full model. When we refer to Table 2, we can see the results that the test accuracy is following the order of the model size. Therefore, a possible explanation for the phenomenon in the CV datasets is that the proposed submodel extraction has found the best mask for the smaller-size models but yet for the full-size.

**Comparison with other baselines.** In most of the cases, the proposed submodel extraction achieves better performance than the existing works, except in a case where the model is trained with CIFAR-100 and updated the parameters using HeteroFL. As we discussed in the last paragraph, an optimal submodel extraction is yet to be found for the full model size. However, in that case, the model with $1./4$ parameters of FedMAP achieves better accuracy than the model with full parameters of FedRolex. Put aside the special case, our proposed submodel extraction can achieve up to 11% accuracy improvement under the same dataset and the same model update approach.

## 5.2 Ablation Study: Two Submodel Extraction Strategies of FedMAP

We conducted experiments involving two strategies of FedMAP extracting the submodels: TopK and Sampling. In the TopK method, we selected parameters with the $K$ largest values of probabilistic masks, while in the Sampling method, parameters are sampled with randomness based on mask probabilities. As depicted in Figure 2, it is evident that TopK outperforms Sampling with a clear margin. This discrepancy in performance may stem from Sampling introducing redundant parameters, an issue not encountered with the TopK method. Notably, both methods exhibited an inflection point as the model size increased, indicating the presence of redundant parameters in the scope of the whole network. Moreover, our findings suggest that our algorithm is capable of accommodating arbitrary computational constraints. This capability allows it to process models of sizes not appear during training, and its performance falls within the range observed between two consecutive training model sizes.

## 6 Conclusion

This work is motivated by a discovery that the existing works cannot find the optimal submodel architectures such that the performance hardly achieves the global optimal. Therefore, we propose FedMAP to utilize a probabilistic mask for optimal submodel architectures. In specific, the probabilistic mask is jointly trained by the FL clients and updated via a quasi-bayesian approach. Upon the approach, we introduce a coordinate descent approach, where the algorithm can alternatively update the mask and the model parameters. Extensive experiments verify the effectiveness of the proposed FedMAP when incorporating the state-of-the-art model to update model parameters.

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

## A  ALGORITHMS FOR FEDMAP WITH MODEL PARAMETERS UPDATE

---

**Algorithm 2** MaskGeneration

---

**Input:** Probabilistic mask $p \in [0,1]^d$, target model size $\gamma \in (0,1]$.

1: Initialize the mask $\mathcal{M} = \{1\}^d$
2: Initialize the sum of mask $m = \{0\}^d$
3: **while** $\|\mathcal{M}\|_1 > \gamma d$ **do**
4:     $\mathcal{M} \sim \text{Bern}(p \odot \mathcal{M})$
5:     $m = m + \mathcal{M}$
6: **end while**
7: **return** $\mathcal{M}, m$

---

**Algorithm 3** FedMAP with Model Updates

---

**Input:** masking learning rate $\eta_m$, learning rate for model update $\eta$, global learning rate for model update $\eta_s$, local updates $K$, Reset Prior $\tau$, Bata distribution parameters $(\alpha_0, \beta_0)$, initial model $\tilde{x}$, loop $\tau$.

1: **for** $T = 0, \tau, 2\tau, \ldots$ **do**
2:     **if** $(T/\tau)\%2 == 0$ **then**                ▷ Obtain the probabilistic mask
3:         $p =$FedMAP$(\eta_m, K, \tau, (\alpha_0, \beta_0), \tilde{x}, \tau)$
4:     **else**                ▷ Update the model parameters
5:         Sample clients $\mathcal{A} \subseteq [N]$
6:         Generate a sequence of masks $\mathcal{M}_i = $ MaskGeneration$(p, \gamma_i)$ for clients $i \in \mathcal{A}$
7:         Communicate the model $\tilde{x} \odot \mathcal{M}_i$ with clients $i \in \mathcal{A}$
8:         **for** $i \in \mathcal{A}$ **in parallel do**
9:             Initialize $x_{t,0}^{(i)} = \tilde{x} \odot \mathcal{M}_i$
10:            **for** $k = 0, \ldots, K-1$ **do**
11:                $x_{t,k+1}^{(i)} = x_{t,k}^{(i)} - \eta \nabla F_i(x_{t,k}^{(i)} \odot \mathcal{M}_i) \odot \mathcal{M}_i$
12:            **end for**
13:            $\Delta x_t^{(i)} = \tilde{x}_t - x_{t,K}^{(i)}$
14:        **end for**
15:        $\tilde{x} = \tilde{x} - \eta_s Agg_{i \in \mathcal{A}}(\Delta x_t^{(i)})$                ▷ As defined in Section 3
16:    **end if**
17: **end for**

---

## B  EXPLANATION FOR PROBABILISTIC MASK UPDATE (PROPOSITION 4.1)

Let $p_i = \sigma(s_i)$. Based on the straight-through estimator, we have

$$\frac{\partial Bern(p_i)}{\partial s_i} = \sigma(s_i)(1 + \exp(-s_i))^{-2} \cdot \exp(-s_i) \tag{10}$$

As our method is equivalent to taking multiple Bernoulli sampling, we have the following conclusion:

$$\frac{\partial f(\mathcal{M}_i|p)}{\partial p} = \begin{cases} 0, & m < \|\mathcal{M}_i\|_\infty - 1 \\ \left(p - p^{2m+1}\right)\left(1 - p^2\right)^{-1}, & \text{Otherwise} \end{cases} \tag{11}$$

## C  PROOF OF THEOREM 3.3

**Lemma C.1.** *Let $\varepsilon = \{\varepsilon_1, \ldots, \varepsilon_a\}$ be the set of random variables in $\mathbb{R}^{a \times d}$. Every element in $\varepsilon$ is independent with others. For $i \in \{1, \ldots, a\}$, the value for $\varepsilon_i$ follows the setting below:*

$$\varepsilon_i = \begin{cases} e_i, & \text{probability} = q \\ \mathbf{0}, & \text{otherwise} \end{cases} \tag{12}$$

*where $q$ is a constant real number between 0 and 1, i.e., $q \in [0, 1]$. Let $|\cdot|$ indicate the length of a set, $\varepsilon \setminus \{\mathbf{0}\}$ represent a set in which an element is in $\varepsilon$ but not $\mathbf{0}$. Then, there is a probability of $(1-q)^a$ for $|\varepsilon \setminus \{\mathbf{0}\}| = 0$, let $avg(\varepsilon)$ be the averaged result with the exception of zero vectors, i.e.,*

$$avg(\varepsilon) = \begin{cases} \frac{1}{|\varepsilon \setminus \{\mathbf{0}\}|} \sum_{i=1}^a \varepsilon_i, & |\varepsilon \setminus \{\mathbf{0}\}| \neq 0 \\ 0, & |\varepsilon \setminus \{\mathbf{0}\}| = 0 \end{cases} \tag{13}$$

*Then, the following formulas hold for $\mathbb{E}\left(avg(\varepsilon)\right)$ and its second norm $\mathbb{E}\|avg(\varepsilon)\|_2^2$:*

$$\mathbb{E}\left(avg(\varepsilon)\right) = (1 - (1-q)^a) \cdot \frac{1}{a} \sum_{i=1}^a e_i; \quad \mathbb{E}\|avg(\varepsilon)\|_2^2 \leq (1 - (1-q)^a) \cdot \frac{1}{a} \sum_{i=1}^a \|e_i\|_2^2 \tag{14}$$

*Proof.* When $q = 0$, the formulas in Equation 14 obviously hold because $\mathbb{E}\left(avg(\varepsilon)\right) = 0$ and $\mathbb{E}\|avg(\varepsilon)\|_2^2 = 0$. As for $q = 1$, since $avg(\varepsilon) = \frac{1}{a} \sum_{i=1}^a e_i$, we leverage Cauchy–Schwarz inequality and get $\mathbb{E}\|avg(\varepsilon)\|_2^2 = \|\frac{1}{a} \sum_{i=1}^a e_i\|_2^2 \leq \frac{1}{a} \sum_{i=1}^a \|e_i\|_2^2$, which is consistent with the formulas in Equation 14. In addition to the preceding cases, we consider some general cases for the probability $q$ within 0 and 1, i.e., $q \in (0, 1)$.

Firstly, we show the proof details for $\mathbb{E}\left(avg(\varepsilon)\right)$. For all $i$ in $\{1, \ldots, a\}$, given that $\varepsilon_i$ is not a zero vector, the coefficient of $e_i$ is based on the binomial distribution on how many non-zero elements in the set $\{\varepsilon_1, \ldots, \varepsilon_{i-1}\} \cup \{\varepsilon_{i+1}, \ldots, \varepsilon_a\}$. Therefore, with the probability $q$ that $\varepsilon_i$ is equal to $e_i$, the coefficient of $e_i$ in the expected form is

$$q \left( \frac{1}{a} \cdot \underbrace{\binom{a-1}{a-1} q^{a-1}}_{(a-1) \text{ non-zero elements}} + \cdots + \frac{1}{1} \cdot \underbrace{\binom{a-1}{0}(1-q)^{a-1}}_{0 \text{ non-zero element}} \right)$$

Then, the coefficient of $\frac{1}{a} e_i$ can be expressed and simplified for

$$q\left( \frac{a}{a} \cdot \binom{a-1}{a-1} q^{a-1} + \cdots + \frac{a}{1} \cdot \binom{a-1}{0}(1-q)^{a-1} \right) \tag{15}$$

$$= q\left( \binom{a}{a} q^{a-1} + \cdots + \binom{a}{1}(1-q)^{a-1} \right) \tag{16}$$

$$= \binom{a}{a} q^a + \cdots + \binom{a}{1} q(1-q)^{a-1} \tag{17}$$

$$= 1 - (1-q)^a \tag{18}$$

where Equation (17) follows

$$\binom{\alpha}{\beta} = \frac{\alpha}{\beta} \cdot \frac{(\alpha-1) \times \cdots \times (\alpha - \beta + 1)}{1 \times \cdots \times (\beta - 1)} = \frac{\alpha}{\beta} \binom{\alpha-1}{\beta-1}, \quad \forall \alpha \geq \beta > 0$$

and Equation (18) follows

$$(q + (1-q))^a = \binom{a}{a} q^a + \cdots + \binom{a}{0}(1-q)^a.$$

Thus, the equation $\mathbb{E}\left(avg(\varepsilon)\right) = (1 - (1-q)^a) \cdot \frac{1}{a} \sum_{i=1}^a e_i$ holds.

Secondly, we provide the analysis for $\mathbb{E}\|avg(\varepsilon)\|_2^2$. Based on the definition for $avg(\varepsilon)$ in Equation (13), we discuss the case $|\varepsilon \setminus \{\mathbf{0}\}| \neq 0$. By means of Cauchy-Schwarz inequality, we can obtain the following inequality:

$$\left\| \frac{1}{|\varepsilon \setminus \{\mathbf{0}\}|} \sum_{i=1}^a \varepsilon_i \right\|_2^2 = \left\| \frac{1}{|\varepsilon \setminus \{\mathbf{0}\}|} \sum_{i, \varepsilon_i \neq \mathbf{0}} \varepsilon_i \right\|_2^2 \leq \frac{1}{|\varepsilon \setminus \{\mathbf{0}\}|} \sum_{i, \varepsilon_i \neq \mathbf{0}} \|\varepsilon_i\|_2^2 = \frac{1}{|\varepsilon \setminus \{\mathbf{0}\}|} \sum_{i=1}^a \|\varepsilon_i\|_2^2 \tag{19}$$

Therefore,

$$\|avg(\varepsilon)\|_2^2 \leq \begin{cases} \frac{1}{|\varepsilon \setminus \{\mathbf{0}\}|} \sum_{i=1}^a \|\varepsilon_i\|_2^2, & |\varepsilon \setminus \{\mathbf{0}\}| \neq 0 \\ 0, & |\varepsilon \setminus \{\mathbf{0}\}| = 0 \end{cases} \tag{20}$$

Apparently, Equation (20) is very similar to Equation (13) in terms of the expression. As a result, we can adopt the same proof framework in the analysis of $\mathbb{E}(avg(\varepsilon))$. Then, we can directly draw a conclusion $\mathbb{E} \|avg(\varepsilon)\|_2^2 \leq (1 - (1-q)^a) \cdot \frac{1}{a} \sum_{i=1}^a \|e_i\|_2^2$. $\qquad\square$

Now, let us consider how to prove the convergence rate. According to Assumption 3.1, the objective function satisfies $L$-smooth, and therefore,

$$\mathbb{E}F(\tilde{\boldsymbol{x}}_{t+1}, \mathcal{M}_*) - F(\tilde{\boldsymbol{x}}_t, \mathcal{M}_*) \tag{21}$$

$$\leq \mathbb{E} \langle \nabla F(\tilde{\boldsymbol{x}}_t, \mathcal{M}_*), (\tilde{\boldsymbol{x}}_{t+1} - \tilde{\boldsymbol{x}}_t) \odot \mathcal{M}_* \rangle + \frac{L}{2} \|(\tilde{\boldsymbol{x}}_{t+1} - \tilde{\boldsymbol{x}}_t) \odot \mathcal{M}_*\|_2^2 \tag{22}$$

$$\leq \mathbb{E} \left\langle \nabla F(\tilde{\boldsymbol{x}}_t, \mathcal{M}_*), -\eta_s \eta C_{\mathcal{M}_t} Agg_{j \in \mathcal{A}} \left( \sum_{k=0}^{K-1} \nabla F_i \left( \boldsymbol{x}_{t,k}^{(i)} \odot \mathcal{M}_i^t \right) \odot \mathcal{M}_i^t \right) \right\rangle + \frac{L}{2} \|(\tilde{\boldsymbol{x}}_{t+1} - \tilde{\boldsymbol{x}}_t) \odot \mathcal{M}_*\|_2^2 \tag{23}$$

According to the lemma above, we have the $j$-th element of $C_{\mathcal{M}_t}$ is

$$(C_{\mathcal{M}_t})_j = \frac{\binom{N}{A} - \binom{N-N_j}{A}}{\binom{N}{A}} \tag{24}$$

In the theorem, we define

$$(\eta_s)_j = \frac{\binom{N}{A}}{\binom{N}{A} - \binom{N-N_j}{A}} \tag{25}$$

The first term of Eq. (23) is

$$\mathbb{E} \left\langle \nabla F(\tilde{\boldsymbol{x}}_t, \mathcal{M}_*), -\eta_s \eta C_{\mathcal{M}_t} Agg_{j \in \mathcal{A}} \left( \sum_{k=0}^{K-1} \nabla F_i \left( \boldsymbol{x}_{t,k}^{(i)} \odot \mathcal{M}_i^t \right) \odot \mathcal{M}_i^t \right) \right\rangle \tag{26}$$

$$\leq -\eta K \mathbb{E} \left\langle \nabla F(\tilde{\boldsymbol{x}}_t, \mathcal{M}_*), Agg_{j \in \mathcal{A}} \left( \frac{1}{K} \sum_{k=0}^{K-1} \nabla F_i \left( \boldsymbol{x}_{t,k}^{(i)} \odot \mathcal{M}_i^t \right) \odot \mathcal{M}_i^t \right) \right\rangle \tag{27}$$

$$\leq -\eta K \mathbb{E} \langle \nabla F(\tilde{\boldsymbol{x}}_t, \mathcal{M}_*), \nabla F(\tilde{\boldsymbol{x}}_t, \mathcal{M}_t) \rangle \tag{28}$$

$$+ \eta K \|\nabla F(\tilde{\boldsymbol{x}}_t, \mathcal{M}_*)\| \cdot \left\| Agg_{i \in [N]} \left( \frac{1}{K} \sum_{k=0}^{K-1} \nabla F_i \left( \boldsymbol{x}_{t,k}^{(i)} \odot \mathcal{M}_i^t \right) \odot \mathcal{M}_i^t - \nabla F_i(\tilde{\boldsymbol{x}}_t \odot \mathcal{M}_i^t) \odot \mathcal{M}_i^t \right) \right\| \tag{29}$$

According to SCAFFOLD (Karimireddy et al., 2020), we have

$$\left\| Agg_{i \in [N]} \left( \frac{1}{K} \sum_{k=0}^{K-1} \nabla F_i \left( \boldsymbol{x}_{t,k}^{(i)} \odot \mathcal{M}_i^t \right) \odot \mathcal{M}_i^t - \nabla F_i(\tilde{\boldsymbol{x}}_t \odot \mathcal{M}_i^t) \odot \mathcal{M}_i^t \right) \right\| \leq \sqrt{6} \eta K G \tag{30}$$

Therefore,

$$\mathbb{E} \left\langle \nabla F(\tilde{\boldsymbol{x}}_t, \mathcal{M}_*), -\eta_s \eta C_{\mathcal{M}_t} Agg_{j \in \mathcal{A}} \left( \sum_{k=0}^{K-1} \nabla F_i \left( \boldsymbol{x}_{t,k}^{(i)} \odot \mathcal{M}_i^t \right) \odot \mathcal{M}_i^t \right) \right\rangle \tag{31}$$

$$\leq -\frac{\eta K}{2} \|\nabla F(\tilde{\boldsymbol{x}}_t, \mathcal{M}_*)\|_2^2 - \frac{\eta K}{2} \|\nabla F(\tilde{\boldsymbol{x}}_t, \mathcal{M}_t)\|_2^2 \tag{32}$$

$$+ \frac{\eta K}{2} \left\| \left( \nabla F(\tilde{\boldsymbol{x}}_t, \mathcal{M}^t) - \nabla F(\tilde{\boldsymbol{x}}_t, \mathcal{M}_*) \right) \cdot \mathbf{1}_{\{\nabla F(\tilde{\boldsymbol{x}}_t, \mathcal{M}_*) \neq 0\}} \right\|_2^2 + \sqrt{6} \eta^2 K^2 G^2 \tag{33}$$

According to SCAFFOLD (Karimireddy et al., 2020) and Assumption 3.2, we have

$$\|(\tilde{\boldsymbol{x}}_{t+1} - \tilde{\boldsymbol{x}}_t) \odot \mathcal{M}_*\|_2^2 \leq O(\eta^2 K^2 G^2) \tag{34}$$

Therefore, we can have the desired conclusion as what Theorem 3.3 presents.

