# OpenReview forum: "One Training Fits All: Addressing Model-Heterogeneity Federated Learning via Architecture Probing"
_ICLR.cc/2024/Conference — ICLR 2024 Conference Withdrawn Submission_

### Official Review · Reviewer_LZmJ · 2023-10-24

**Soundness:** 2 fair
**Presentation:** 3 good
**Contribution:** 2 fair
**Rating:** 3
**Confidence:** 5

**Summary:**

This paper proposes a heterogeneous model architecture federated learning approach (FedMAP) based on mask probing to find best neural architecture for local clients in FL-edge scenarios.
Specifically, the paper introduced a mask architecture probing method to prune out unimportant weight parameters in a given large global model and search for optimal submodels for local clients.
Hence, FedMAP is able to perform post-training model deployment for newly joined clients based on their local resources.
Additionally, this paper provides a theoretical analysis of heterogeneous FL.

**Strengths:**

**1.** This paper provides theoretical analysis of the convergence of collaboratively training architecture heterogeneous sub-models in federated learning.

**2.** The proposed FedMAP enables post-training model deployment for newly joined FL clients

**Weaknesses:**

**1.**  Experiment results look weak. For instance, the final model performance trained by FedMAP is around 30% accuracy, and the model doesn't look like *converged*.

**2.**  The author mentioned post-training deployment for newly joined clients. However, I didn't see it in experiments. Specifically, after FL training finished, I expected to see the performance evaluation for different submodels with different resource capacities.


**3.** Search for masks during FL may introduce extra **computational** and **communication** costs; however, it has not been discussed in the paper.

**4.** Based on my understanding, applying a mask to the model when performing local training, the computational resource reduction at the edge is not significant. Masking might not shrink the actual model size, the activations and gradients of masked over still zero paddings in memory.

**Questions:**

Please kindly address the points I proposed in Weaknesses section.

Overall, the experiments look not strong enough in this paper and cannot support the arguments they proposed in the paper. If authors could improve their experiments and provide more persuasive results,  I'll change my mind.
For instance, the model performance achieved by FedMAP looks divergence in both datasets, which can not support their convergence analysis. Authors may use IID FL FL settings to achieve convergences performance and further support their convergence analysis.

Also, as mentioned the post-training deployment, after FL training is finished, the submodels performance with different capacities should also been discussed in experiments sections.

---

### Official Review · Reviewer_VuHT · 2023-10-29

**Soundness:** 3 good
**Presentation:** 3 good
**Contribution:** 2 fair
**Rating:** 5
**Confidence:** 4

**Summary:**

This paper proposes to address the problem of model heterogeneity in federated learning through a novel approach called FedMAP. FedMAP enables the dynamic creation of customized model sizes aligned with the client's budget, thereby improving the performance of state-of-the-art approaches to heterogeneous model updates and submodel extraction. Through experiments, the authors demonstrate the effectiveness of FedMAP in achieving exceptional performance in federated learning.

**Strengths:**

1. The paper is well-organized and clearly presents the proposed approach, FedMAP, in a concise and easy-to-understand manner. The authors provide clear explanations of the theoretical concepts and experimental results, making it easy for readers to follow along.

2. The paper provides the first theoretical analysis on the convergence of federated learning algorithms under model heterogeneity. The analysis reveals the limitations of prior works and highlights the importance of optimizing submodel architectures to mitigate performance degradation.

3. Extensive experiments. The authors conduct extensive experiments on two vision benchmark datasets and one natural language processing dataset to evaluate the effectiveness of FedMAP. The results demonstrate that FedMAP outperforms state-of-the-art approaches to heterogeneous model updates and submodel extraction, achieving exceptional performance in federated learning.

**Weaknesses:**

1. The idea of model masking strategy is more like the neural architecture search strategy for FL where the network architecture can also flexibly change for each client. It is better to conduct a literature review on those works and present a comparison between this work and those works.

2. Vague motivation for the theoretical analysis on model architecture. While the paper provides a sound theoretical analysis on the convergence of federated learning algorithms under model heterogeneity, the authors do not provide a clear motivation for why this analysis is necessary or how it contributes to the proposed approach, FedMAP.

3. Lack of discussion on practical limitations. While the authors demonstrate the effectiveness of FedMAP in improving federated learning performance, they do not discuss the practical limitations of their approach. For example, it is unclear how FedMAP would perform in scenarios with large-size models, e.g., large language models. A discussion of these limitations would provide a more complete picture of the approach's strengths and weaknesses.

**Questions:**

None.

---

### Official Review · Reviewer_hWhP · 2023-10-31

**Soundness:** 3 good
**Presentation:** 3 good
**Contribution:** 3 good
**Rating:** 5
**Confidence:** 4

**Summary:**

This paper proposes a novel approach called FedMAP to address model-heterogeneity in federated learning through architecture probing. FedMAP enables the dynamic creation of customized model sizes aligned with the client's budget, and through experiments, it demonstrates the effectiveness of FedMAP in improving state-of-the-art approaches to heterogeneous model updates and extracting submodels with exceptional performance. The paper also provides a theoretical analysis of the convergence of federated learning algorithms under model heterogeneity.

**Strengths:**

1. The paper provides a theoretical analysis of the convergence of federated learning algorithms under model heterogeneity.
  2. The results show that FedMAP significantly outperforms existing approaches, particularly for low-resource clients.

**Weaknesses:**

1. This method bears some resemblance to FL methods with dropout[1,2] (excluding Fjord), even incorporating a trainable mask component. The authors should investigate these papers and compare them with FedMAP as baselines.

2. Why is DepthFL mentioned in the Introduction but not compared in the Experiments section?

3. It would be helpful to include an illustration demonstrating how FedMAP extracts submodels and aggregation models.

4. Must the proportions of the full model, such as {1, 1/4, 1/16, 1/64}, be predefined?

[1] Expanding the Reach of Federated Learning by Reducing Client Resource Requirements.

[2] Adaptive Federated Dropout: Improving Communication Efficiency and Generalization for Federated Learning.

**Questions:**

See Weaknesses

---

### Official Review · Reviewer_UbYG · 2023-11-01

**Soundness:** 2 fair
**Presentation:** 2 fair
**Contribution:** 2 fair
**Rating:** 3
**Confidence:** 3

**Summary:**

The paper addresses the issue of systems heterogeneity in federated learning (FL) and focuses on model heterogeneity, an effective strategy to deal with limited and different local computational capacities. This work argues how the extraction of appropriate submodels by current approaches is highly architecture-dependent and easily suffers from degradation. To this end, FedMAP (Federated Learning with Masked Architecture Pruning) is introduced. FedMAP leverages a probabilistic mask to select the important part of the network and explicitly optimizes submodel architectures, while allowing the generation of new submodels for potentially new clients.
Theoretical findings validate the paper’s claims. Empirical evaluation is run on FL benchmark datasets, showing the efficacy of the proposed approach.

**Strengths:**

- The paper addresses a relevant issue for the FL community, i.e. model heterogeneity for handling different local computation capabilities
- Theoretical proofs and analyses validate the paper’s claims. The paper introduces an analysis on the convergence of FL under model heterogeneity.
- FedMAP is empirically validated on different tasks (image classification and next character prediction) against state-of-the-art (sota) approaches. The results obtained with FedMAP exhibit significant gains w.r.t. to the baselines.
- Realistic federated settings in terms of number of clients, architectures deployed, and addressed tasks.

**Weaknesses:**

- Some relevant related works are not discussed. For instance, how does the model-heterogeneity approach relate with the Lottery Ticket Hypothesis [1] literature? And model sparsity? How does this work relate with centralized approaches aiming at finding subnetworks showing high performance e.g. [2][3]?
- It is not clear how much FedMAP impacts the local computation on the resource-constrained devices w.r.t. more simple and standard baselines, e.g. FedAvg.
- Necessary details for reproducing the results are not provided, e.g. local training hyperparameters, number of communication rounds, clients participation. Are the experiments run under partial or full clients’ participation? How does FedMAP behave when faced with different levels of clients' participation?
- To the best of my understanding, the experiments do not analyze the behavior of FedMAP when changing $\gamma_i$, which measures the local available resources and is left constant across clients. This implies the results in Table 1 and 2 represent a simplified setting w.r.t. the one FedMAP was introduced for.
- Limitations are not explicitly addressed. For instance, masking out some parts of the network may result in poor performance towards minority groups, with consequent fairness concerns.
- Even if the presented results are promising, the experiments are not run on large-scale federated datasets (e.g., Landmarks-Users-160k, FEMNIST), they are limited to one single setting per dataset (e.g., partial vs full participation is not analyzed, as well as the impact of data heterogeneity on the results) and the comparison with the state of the art is limited. Possible additions are [4][5][6]. In addition, settings relevant to FedMAP (e.g., clients having different local resources) are not analyzed.


**Typos**
- pg 4 under “global model aggregation”: the sentence should probably be “the server collects” and not “the clients collect”
- Some unnecessary capital case letters all over the paper, e.g. pg. 5 under “solution overview” → “We”

**References**
[1] Frankle, Jonathan, and Michael Carbin. "The lottery ticket hypothesis: Finding sparse, trainable neural networks." ICLR (2019).
[2] Hattie Zhou, Janice Lan, Rosanne Liu, and Jason Yosinski. Deconstructing lottery tickets: Zeros, signs, and the supermask. Advances in neural information processing systems, 32, 2019.
[3] Vivek Ramanujan, Mitchell Wortsman, Aniruddha Kembhavi, Ali Farhadi, and Mohammad Rastegari. What’s hidden in a randomly weighted neural network? In Proceedings of the IEEE/CVF Conference on Computer Vision and Pattern Recognition, pp. 11893–11902, 2020.
[4] Samuel Horvath, Stefanos Laskaridis, Mario Almeida, Ilias Leontiadis, Stylianos I. Venieris, and Nicholas D. Lane. FjORD: Fair and accurate federated learning under heterogeneous targets with ordered dropout. CoRR, abs/2102.13451, 2021
[5] Liao, Dongping, et al. "Adaptive Channel Sparsity for Federated Learning Under System Heterogeneity." Proceedings of the IEEE/CVF Conference on Computer Vision and Pattern Recognition. 2023.
[6] Isik, Berivan, et al. "Sparse random networks for communication-efficient federated learning." ICLR (2023).

**Questions:**

1. Does FedMAP increase the communication cost between clients and server w.r.t. the baselines?
2. How does the computation linked with FedMAP increase the computational local costs?
3. Results are introduced on datasets partitioned according to the Dirichlet distribution with $\alpha=0.3$. What happens when the scenario is more heavily heterogeneous (e.g., $\alpha=0.1$)? And how does the behavior of FedMAP change when faced with homogeneous scenarios (i.e, $\alpha \rightarrow \infty$)?
4. Do the authors have an intuition on why the submodel extraction does not achieve the best performance when handling the full model size?
5. How is the proposed method affected by the distribution of $\gamma_i$ across clients?
6. Communication is a relevant aspect when deploying FL algorithms. How does FedMAP compare with the state of the art in terms of speed of convergence, i.e. number of rounds to reach a target accuracy?
7. How does the masking process obtained with FedMAP compare with the other introduced approaches? How do the local subnetworks compare?

---

### Public Comment · ~Saurav_Prakash1 · 2023-11-21
**Closely related work**

Dear Authors,


I found your work to be interesting. In this regard, I would also like to draw your attention to our following related work, where we proposed PriSM methodology to extract small sub-models, while maintaining a near full coverage of the FL server model. As we show in the paper, we perform significantly better in comparison to existing SOTA, particularly in the regime when each client is extremely resource constrained. It would be good to see comparisons of your approach with PriSM.


* Niu, Y., Prakash, S., Kundu, S., Lee, S. and Avestimehr, S., 2023. "Overcoming Resource Constraints in Federated Learning: Large Models Can Be Trained with only Weak Clients". Published in Transactions on Machine Learning Research, 2023 (link: https://openreview.net/forum?id=lx1WnkL9fk). Partly presented at the NeurIPS Workshop on Recent Advances and New Challenges in Federated Learning (FL-NeurIPS), 2022 (link: https://openreview.net/forum?id=e97uuEXkSii).


Thanks and Regards,


Saurav